# Plasmon Tuning of Liquid Gallium Nanoparticles through Surface Anodization

**DOI:** 10.3390/ma15062145

**Published:** 2022-03-15

**Authors:** Chih-Yao Chen, Ching-Yun Chien, Chih-Ming Wang, Rong-Sheng Lin, I-Chen Chen

**Affiliations:** 1Institute of Materials Science and Engineering, National Central University, Zhongli 320, Taiwan; 102389004@cc.ncu.edu.tw (C.-Y.C.); 109329005@cc.ncu.edu.tw (C.-Y.C.); 2Department of Optics and Photonics, National Central University, Zhongli 320, Taiwan; cmwang@dop.ncu.edu.tw (C.-M.W.); 109226042@cc.ncu.edu.tw (R.-S.L.)

**Keywords:** tunable plasmon resonance, liquid gallium nanoparticle, surface anodization, shape deformation

## Abstract

In this work, tunable plasmonic liquid gallium nanoparticles (Ga NPs) were prepared through surface anodizing of the particles. Shape deformation of the Ga NPs accompanied with dimpled surface topographies could be induced during electrochemical anodization, and the formation of the anodic oxide shell helps maintain the resulting change in the particle shape. The nanoscale dimple-like textures led to changes in the localized surface plasmon resonance (LSPR) wavelength. A maximal LSPR red-shift of ~77 nm was preliminarily achieved using an anodization voltage of 0.7 V. The experimental results showed that an increase in the oxide shell thickness yielded a negligible difference in the observed LSPR, and finite-difference time-domain (FDTD) simulations also suggested that the LSPR tunability was primarily determined by the shape of the deformed particles. The extent of particle deformation could be adjusted in a very short period of anodization time (~7 s), which offers an efficient way to tune the LSPR response of Ga NPs.

## 1. Introduction

Plasmonics has emerged and drawn much attention over the past decade due to its capabilities for the manipulation of electromagnetic radiation and enhancement of light–matter interactions [1,2,3,4,5]. Among the various plasmonic nanostructures, the simple fabrication routes and unique optical properties of metal nanoparticles (NPs) make them potentially useful in diverse fields [6,7,8,9,10]. The noble metals gold (Au) and silver (Ag) have been the most popular materials used in plasmonics. However, localized surface plasmon resonances (LSPRs) arising in these NPs are basically restricted to the visible or near infrared spectral region because of the inherent limitations associated with their electronic structures [11]. Recently, liquid gallium (Ga) has emerged as an alternative plasmonic material since it exhibits nearly free electron-like Drude behavior over a broad bandwidth from the ultraviolet to the infrared region [12]. In addition, liquid Ga NPs possess attractive properties including excellent chemical stability by the self-terminating native oxide shell and ease of processing by various approaches [13], making them promising candidates for different applications such as surface-enhanced Raman scattering (SERS) [14,15], solid–liquid phase change memories [16], waveguiding [17], ellipsometric biosensing [18,19], and plasmon-based catalysis [20].

The LSPR wavelength of metal NPs has a strong dependence on the particle shape and size [21,22]. Various approaches have been explored to synthesize metal NPs with different morphologies [22,23,24,25]. Nevertheless, it is a more direct and intuitive way to tune the LSPR wavelength of metal NPs by changing their shape. Chemical etching and mechanical compression have been used to reshape Au or Ag NPs and tune the wavelengths of LSPR spectra [26,27,28]. However, the etch technique will cause a significant reduction in both the particle volume and resonance peak intensity, whereas high pressure is required to result in plastic deformation of solid metal NPs by anisotropic compression. Due to its liquid phase, Ga can be readily manipulated and deformed at room temperature.

In this study, tunable plasmonic Ga NPs on Si substrates were realized via surface anodization. The stress generated during surface oxidation promotes uneven deformation of Ga NPs, leading to dimple-like textures and thus tuning the LSPR response. Reshaping of Ga NPs could be induced with an ultrathin oxide overlayer in a very short anodization time. Moreover, the optical properties of the Ga NPs were investigated as a function of anodization time. The experimental data, along with finite-difference time-domain (FDTD) simulations, suggest that the dramatic LSPR shift is caused by electrochemical deformation of the nanoparticles.

## 2. Materials and Methods

N-type Si (100) wafers were used as substrates for the deposition of Ga NPs. Prior to nanoparticle growth, Si wafers were cut into 15 × 25 mm pieces, cleaned with acetone, and then rinsed with deionized water. Ga NP films with a mass thickness of 70 nm were deposited at room temperature in a thermal evaporator. The evaporation rate was fixed at 0.1 nm/s. Due to the high surface energy of gallium metal [29], the growth of Ga films follows the Volmer–Weber growth mode [30] and thus Ga adatoms tend to self-assemble to form discrete nanoparticles on Si substrates.

Surface anodization of Ga NPs was carried out in a stirred solution of 1% H_3_PO_4_ using a conventional two-electrode cell with a Pt counter electrode. During anodization, the electrolyte temperature and the applied voltage were maintained at 30 °C and 0.7 V, separately. The anodization time was systematically varied from 0.2 to 7 s.

The morphology of the as-deposited and surface-anodized Ga NPs were examined using a scanning electron microscope (SEM; Hitachi 8220). X-ray diffraction (XRD) characterization of the samples was analyzed by a Bruker D8 diffractometer (Cu Kα radiation). X-ray photoelectron spectroscopy (XPS; Thermo VG-Scientific Sigma Probe) was performed to characterize the surface chemical composition of the Ga NPs and evaluate the thickness of the oxide shell. A UV–VIS–NIR spectrometer (BWTEK BRC 112E) was used to characterize the Ga NPs through diffuse reflectance spectra by an integrating sphere, and the illustration of the UV–VIS–NIR system configuration for measuring LSPR characteristic is shown in Appendix A.

The finite difference time-domain (FDTD) simulation was performed to model the optical responses of the as-deposited and surface-anodized Ga NPs on Si substrates. Infinite arrays of nanoparticles were considered in the simulations by using the periodic boundary condition in the lateral direction. Perfectly matched layers were used as boundary conditions in the incident direction. The dielectric constants of liquid Ga and solid Ga_2_O_3_ shell were referred from [31,32], respectively, and Palik’s data were used for Si [33]. Under a normal incident broadband source, the simulated reflectance spectra were calculated with a frequency-domain monitor located behind the light source [34]. To save on computational time of the simulations, a mesh dimension of 5 nm was used while the mesh step was reduced to 1 nm around NPs.

## 3. Results and Discussion

A typical SEM image (Figure 1a) of the as-deposited Ga NPs revealed randomly distributed nanoparticles on the substrate. Since diffusion-limited ripening dominates in the ensemble formation at room temperature, an ensemble consisting of larger Ga NPs surrounded by smaller ones was observed (the inset of Figure 1a). Figure 1b shows a histogram of the number-weighted distribution of the particle size from the as-deposited Ga NP sample. As depicted in Figure 1b, particle size distribution could be divided into two groups: one showed a well-defined Gaussian peak with an average diameter of 260.7 ± 61.4 nm and the other corresponded to a significant number of small Ga NPs formed during the coarsening process. Due to size-dependent melting point depression, submicrometer-sized liquid Ga could be undercooled down to −123 °C [35,36]. GI-XRD analysis could provide evidence of supercooling of Ga NPs. From the XRD pattern obtained from the as-deposited Ga NPs (shown in Figure 1c), the appearance of two overlapping broad peaks centered at ~36° and ~44°, which corresponded to the liquid Ga state, may be attributed to the short-range structural order in liquid Ga NPs [37,38]. From the previous study, covalent Ga–Ga bonding in the short-range order clusters would hinder the crystallization of liquid Ga into the ordered solid phase [39]. Therefore, the liquid form of Ga NPs was maintained prior to anodization by means of a supercooled state and self-terminating native oxide shell formed when exposed to the ambient atmosphere. From the XPS spectra, the surface gallium to oxygen ratio of the Ga NPs was nearly unchanged after exposure to the ambient for a month (shown in Appendix A), indicating that the oxide shell could be rapidly formed and fairly dense to prevent further oxidation.

Figure 2a–c shows the representative SEM images of a series of the surface-anodized Ga NPs obtained from different anodization times ranging from 0.2 to 7 s. Compared to the as-deposited Ga NPs with a smooth surface (Figure 1a), the dimples randomly appeared on the surface of the anodized nanoparticles and the progressive dimple deepening with anodization time was obvious. The time for dimple texturing can be greatly reduced by increasing the applied voltage. However, the extent of particle deformation cannot be well-controlled when the voltage is above 1 V. Thus, an applied voltage of 0.7 V was chosen to maintain controllability and avoid a lengthy process time for lower-voltage conditions.

Figure 3a shows the representative Ga 3d XPS spectra of the nanoparticle samples before and after surface anodization. Before anodization, the deconvolution of the Ga 3d signal consisted of three peaks centered around 18.5, 20.1, and 20.7 eV, corresponding to metallic Ga, Ga_2_O and Ga_2_O_3_, respectively (shown in Figure 3b). From previous studies, the chemistry of gallium is dominated by +1 and +3 oxidation states [40,41,42,43]. The +2 state is typically unstable and thus GaO could not be detected [43,44]. Compared to the as-deposited sample, an additional peak (located at ~21.6 eV) corresponding to Ga(OH)_3_ [43] could be observed (shown in Figure 3c). As depicted in Figure 3, the intensity of the oxide peak became more pronounced after anodization, revealing the growth of the oxide shell during anodic oxidation.

In order to clarify the effect of the anodic gallium oxide shell on the properties of Ga NPs, the shell thickness was evaluated by quantitative analysis of the XPS spectra using a formulaic method developed by Shard et al. [45]. The detailed calculation procedures can be found in [45]. The Ga 3d level was chosen due to the higher kinetic energy of the emitted photoelectrons, which results in a larger sampling depth [46]. The calculated oxide layer thickness as a function of the anodization time is illustrated in Figure 4. It should be mentioned that the as-deposited Ga NP exposed to air would be covered by a native oxide shell with a thickness of ~0.7 nm, determined by the same calculation approach. Within a very short period of time (~0.2 s), as depicted in Figure 4, the transition from a rapid oxidation stage to a much slower growth stage occurred in the thickness ranging between 1.6 and 1.7 nm, which could be defined as the critical thickness and determined by the intersection of the two fitting-lines. Beyond the critical oxide thickness, the oxidizing anions would be limited to migrate from the electrolyte to the Ga/Ga_2_O_3_ interface, leading to a monotonic decrease in growth rate.

To our knowledge, this is the first report describing the surface anodization of liquid Ga NPs, and thus the formation mechanism of surface dimples is not clear. For comparison, thermal oxidation of Ga NPs was carried out in ambient oxygen at 300 °C for 15 min, which resulted in the growth of a 2.31 nm-thick oxide shell. In contrast, dimple morphology appeared on the surface of the anodized Ga NPs with the approximately same oxide thickness (~2.36 nm) as the thermally-oxidized sample (shown in Figure 2d), implying that dimple formation could be attributed to anodization processing. It has been reported that the electrochemical oxidation of metals could generate compressive interfacial stress [47,48]. Due to the liquid nature of gallium, we suggest that morphological instability on the conformal oxide growth may develop through a non-uniform distribution of the electric field in the early stage of anodization, leading to deformation of the liquid nanoparticles and thus the formation of dimpled morphologies on the particle surface. As the anodization time increased from 7 to 12 s, the dimple deepening was retarded by the formation of thicker oxide layers, while nanoporous structures gradually formed on the dimple surface (shown in Appendix A). The nanopore size increased by further increasing the anodization time (Appendix A). After a relatively long period of anodization (~70 s), Ga NPs are completely anodized. In this stage, irregularly-shaped Ga NPs are left on the substrate due to dissolution of nanoporous oxide structures (Appendix A).

Representative reflectance spectra of the surface-anodized Ga NPs are shown in Figure 5, demonstrating a gradual red-shifting of the resonance wavelength with an increase in the anodization time. The plot of the time-dependent LSPR peak shift is also presented in Figure 4. The LSPR shift could reach 77 nm after anodization for 7 s. As shown in Appendix A, the interparticle spacing was insignificantly changed, even after a long period of anodization time. Due to the formation of a gallium silicide layer at the interface [49], the Ga NPs would adhere firmly to the substrate during anodization and thus the interparticle spacing is basically unchanged. We considered that the resonance shift caused by the NP anodization may depend on three factors: size of metallic gallium core, oxide shell thickness, and particle shape. When the anodizing process was carried out under a relatively low applied potential (0.7 V) in a very short time (≤7 s), only a small amount of Ga would be consumed and converted into an ultrathin (≤2.36 nm) oxide layer. Thus, it is expected that a red-shift of the LSPR band with increasing the oxide thickness may compensate for a blue-shift of the LSPR caused by a reduction in the metallic core size. However, further increase in the anodization time would result in the gradual disappearance of the LSPR band since Ga NPs would be gradually and then completely oxidized (shown in Appendix A).

The effect of surface oxidation on the LSPR signatures of Ga NPs was investigated by experiments and theoretical simulation. Thermally-oxidized Ga NPs with an oxide shell thickness ranging between 0.7 nm and 2.36 nm (determined by XPS analysis) were experimentally examined since the effect of dimple morphology on the plasmon resonance can be ruled out for thermally-oxidized nanoparticles with smooth surfaces. For the calculations, the particle was assumed to be with a core-shell hemispherical shape, and its overall diameter was 260 nm while the Ga_2_O_3_ shell thickness varied from 1 nm (native oxide) to 3 nm. As illustrated in Figure 6a, the dip in the calculated reflectance spectra, which corresponded to the in-plane LSPR mode, was nearly fixed at around 460 nm, and this phenomenon could be obtained in the experimental results (shown in Figure 6b). Both theoretical and experimental results suggest that the surface oxidation of the Ga NPs only exerted an insignificant influence, so the oxide layer was disregarded in following simulations.

At the initial stage of anodization (<1.5 s), only very shallow dimples were formed on the particle surface (Figure 2) and thus the shape of anodized Ga NPs was close to that of the as-deposited ones, corresponding to a negligible red-shift (Figure 4). When the anodization time extended to more than 2.5 s, the dimple-textured structure became gradually deeper, causing wall-like and spike shapes. In parallel to the experimental measurements, theoretical calculations were performed to more accurately relate the effect of the shape deformation of Ga NPs on their optical properties. The previous study showed that compressive oxidative stresses to the order of 0.1 GPa were observed during the anodization of aluminum [48]. Since the volumetric change of liquid Ga is small (ΔV/V_0_ < 4%) under uniform compression up to 1 GPa [50], the variation of the nanoparticle volume after anodization could be considered negligible. For the as-deposited Ga NPs, the particle geometry was modeled as a smooth hemisphere with a diameter of 260 nm on top of a 5 μm-thick Si. Dimple-textured hemispheres with three different dimple depths were used for FDTD simulation of Ga NPs anodized for various periods of time, and their structures were obtained by randomly carving out parts of smaller, different-sized spheres from smooth Ga hemispheres with a diameter of 300 nm (shown in Figure 7a). Thus, the volume of the removed part (circular dimple) can be determined by the intersection of two spheres. In addition, a part of the removed volume would be calculated at repetition due to the intersections of three spheres, so a simplified formula was used to calculate the volumes of the intersections of three spheres [51]. In order to avoid the size effect on LSPR shift, the particle volume after the removal process was maintained roughly the same as that of the as-deposited one. Although the extensive deformation may result in slightly narrower gaps among particles, the simulation results showed that the LSPR shift was only ~4 nm as the nanogap spacing changed from 20 nm to 1 nm (Appendix A). We theoretically calculated the reflectance spectra, which are displayed in Figure 7d. From the spectrum calculated for the as-deposited Ga NPs, we found that the resonance wavelength was around 461 nm. Regarding the dimple-textured hemispheres, the wavelength of the LSPR signature would shift to around 473 nm, 516 nm, and 534 nm respectively with various average dimple depths (25 nm, 27 nm, and 31 nm). Although the calculated NP model may not perfectly reproduce the geometry of real anodized Ga NPs, a red-shift in the resonance wavelength was clear for the simulated dimpled texture, which was also in good agreement with the trend in the LSPR shift obtained from the experimental results. The electric field distributions of a smooth hemisphere and a dimple-textured are shown in Figure 7c,d. From Figure 7d, the electric field was concentrated near spikes or narrow walls, and thus the hotspots could clearly be observed [52]. It is suggested that the severe morphological changes of Ga NPs affected the distribution of the electric field, leading to the generation of hotspots and a dramatic increase in LSPR shift.

## 4. Conclusions

In summary, we demonstrated a simple and efficient approach to tune the LSPR response of liquid Ga NPs through surface anodization. Due to compressive stress generated during the formation of the anodic oxide overlayer, the nanoparticle surface could develop dimple-like textures, leading to a red-shift of the LSPR wavelength. This electrochemically driven NP deformation could be induced with an ultrathin oxide layer in the first few seconds of anodic oxidation, and thus the particle volume could approximately remain unchanged. Moreover, the red-shift phenomenon of LSPR response and generation of hot spots by surface anodization could have potential application in optical sensors [53,54]. In addition to experimental measurements, the theoretical simulations also suggest that the formation of dimple textures is the dominant factor that affects the LSPR shift. To the best of our knowledge, this is the first preliminary study of tunable plasmonics based on the shape deformation of liquid nanoparticles. This work may pave the way for broadening the applicability of liquid metals in the field of plasmonics.

## Figures and Tables

**Figure 1 materials-15-02145-f001:**
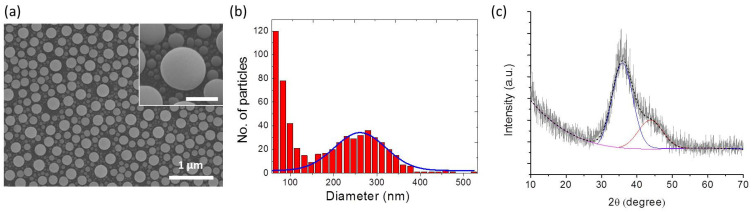
(**a**) Top-view SEM image of the as-deposited Ga NPs on Si. Inset: 30°-tilted-view SEM image of Ga NPs (scale bar: 200 nm). (**b**) Particle size histogram from the SEM graph of Ga NPs. (**c**) XRD pattern of the as-deposited Ga NPs.

**Figure 2 materials-15-02145-f002:**
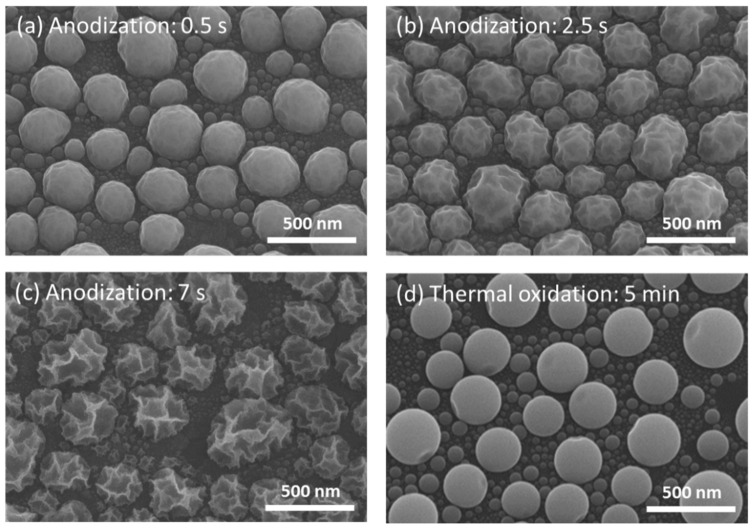
SEM images from a 30°-tilted view of Ga NPs after anodization for (**a**) 0.5 s, (**b**) 2.5 s, (**c**) 7 s, and (**d**) after thermal oxidation at 300 °C for 5 min.

**Figure 3 materials-15-02145-f003:**
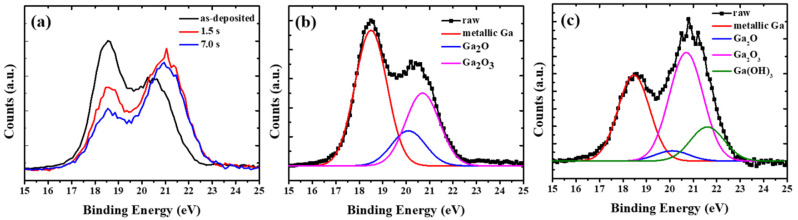
(**a**) XPS spectra of the 3d level of Ga NPs after anodization for different time periods. Deconvolution analysis of XPS spectra of (**b**) the as-deposited and (**c**) anodized samples (anodization time: 7 s), respectively.

**Figure 4 materials-15-02145-f004:**
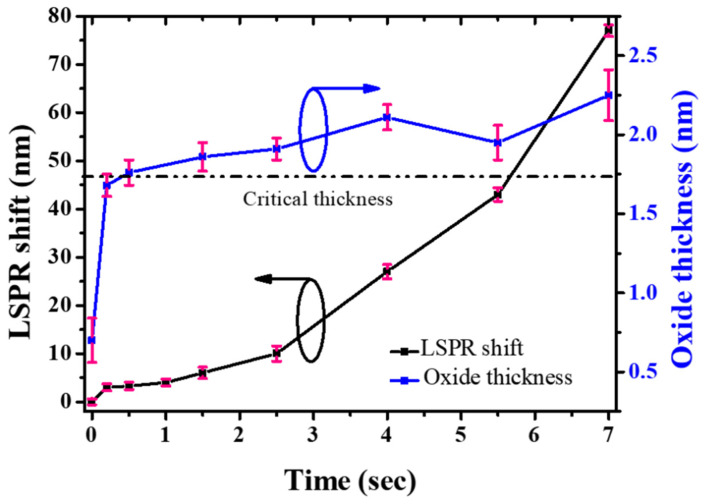
Measured LSPR shift and oxide layer thickness as a function of anodization time.

**Figure 5 materials-15-02145-f005:**
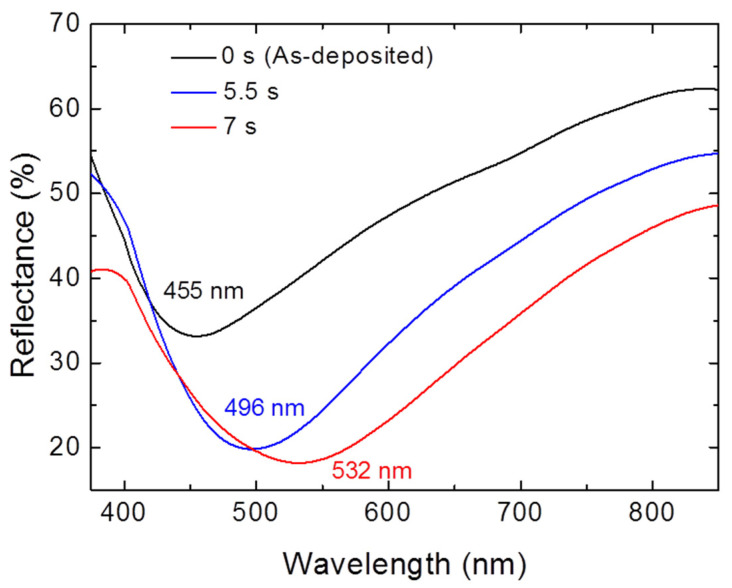
Experimental UV–Visible spectra of representative surface-anodized Ga NPs.

**Figure 6 materials-15-02145-f006:**
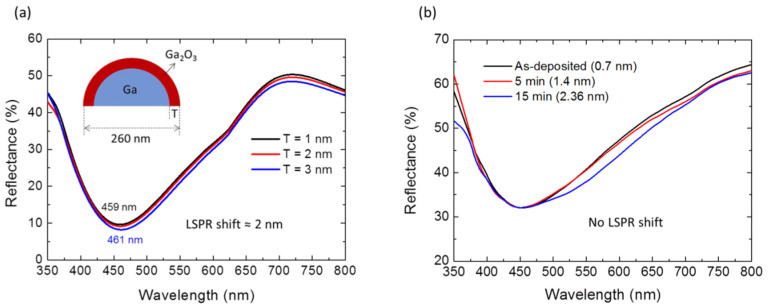
(**a**) FDTD simulations of the effect of surface oxidation on the reflectance spectra of infinite Ga NP arrays. The inset shows a side view of the core-shell nanoparticle. (**b**) Experimental reflectance spectra of Ga NPs after thermal oxidation at 300 °C for different time periods. The corresponding oxide thicknesses calculated from the XPS data are shown in parentheses.

**Figure 7 materials-15-02145-f007:**
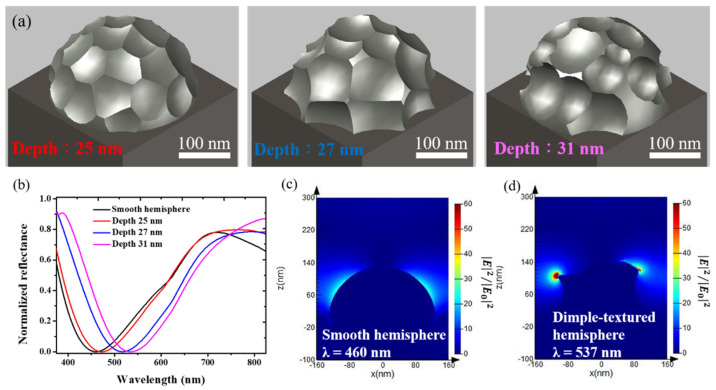
(**a**) Geometries of the dimple-textured hemisphere used for FDTD simulation of anodized Ga NPs with different dimple depths. (**b**) Calculated reflectance spectra for infinite arrays of the smooth and dimple-textured hemispheres, corresponding to the as-deposited and anodized Ga NPs, respectively. Electric field intensity distribution of (**c**) smooth and (**d**) dimple-textured hemispheres. Both output images were normalized to the intensity of the excitation source.

## Data Availability

Not applicable.

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
