# Peer review of "Plasmon Tuning of Liquid Gallium Nanoparticles through Surface Anodization"

_materials, 2022, doi:10.3390/ma15062145_

Round 1

Reviewer 1 Report

In their manuscript, Chen and co-workers discuss the effects of surface anodization on the properties of plasmonic liquid gallium nanoparticles. The process is shown to deform the surface of the particles, and anodization time is tune to shift the plasmon resonance wavelength as a result of the shape deformation, as confirmed by a combination of experiments and numerical results.

Generally, the analysis seems properly framed, the main results of the numerical simulations are in reasonably good agreement with experiments and experimental procedure is well detailed. However, some aspects of the study might be addressed to refine the reported analysis. In particular:

  • I find the way authors are presenting results a bit misleading. As far as I understand, the main mechanism for tuning the plasmonic resonance is the anodization-induced deformation of the particle geometry, as shown by experiments (Fig. 4) and numerical results (Fig. 7). Figure 4 though relates the plasmon frequency shift to the oxide thickness that is generated throughout the anodization process, that however Fig. 6 shows to have negligible impact on the plasmon spectral position. If NP surface deformation is ascertained to be the dominant mechanism of the tuning, I would suggest author to focus on this claim, and, as a further investigation, to discuss the (negligible) effects of oxide thickness on the plasmon frequency and its role to “maintain the resulting change in the particle shape” (abstract, line 12), which remains a bit throughout the manuscript. This could be done by re-arranging paragraphs/figures in the manuscript, or by clarifying the rationale and implications of the investigation.
  • What does “short time” (in the abstract) actually mean? Does it refer to the ~0.2s mentioned in the main text? I would suggest author to specify the value in the abstract/introduction as well.
  • Regarding the FTDT implementation, have the authors treated the Ga NP as a single object, or in array configuration? The boundary conditions of the simulation should be included in the text.
  • Page 7 line 189: could the authors comment on the rationale of considering the volume of the deformed particle approximately equal to the as-deposited case, although having removed Ga from the surface?
  • Have the authors estimated the thickness of native oxide when NPs are exposed to air with the same method introduced at page 3 line 120? And how have they evaluated the oxide critical thickness of 1.7 nm?
  • Is anodization time the only degree of freedom used to induce surface deformation? Does the anodization voltage (here fixed at 0.7V) have any role?
  • Page 4 line 142: according to the caption of Fig.2, panel (d), and not (c) is the thermally-oxidized sample.
  • Could the author comment on the trend observed in Fig.4, showing a saturation regime for the oxide thickness, which does not correspond to the same saturation observed for the LSPR peak? Have the author try anodization for longer times than 7s?
  • Regarding the negligible impact of the shell thickness, how much the dielectric permittivities of Ga and Ga2O3 differ one from each other? Are simulations in Fig.7 disregarding the oxide layer?

Some further minor remarks on the manuscript, to address before resubmission, here below:

  • The manuscript is overall well written, although some misprints and grammar slips are still present. Authors may want to fix them before resubmission;
  • I would suggest authors to modify Fig.4 in order to make it easier to read, e.g. by adding arrows to better associate the curves to the correct vertical axis of the plot.
  • Plot labels are not the same among figures, which should be fixed.
  • A scale bar in Fig.7a should be included.

Thus, in light of those concerns, I am unable to recommend the manuscript for publication in the present form. However, if the points will be addressed by authors, I will be ready to reconsider the work, which may be suitable for publication in Optical Materials.

Reviewer 2 Report

In this manuscript, the authors report on a new method for shaping liquid gallium (Ga) nanoparticles and tuning their plasmonic properties. The method consists in a surface anodization process performed using a N-type silicon (Si) wafer with deposited Ga nanoparticles as an electrode and a platinum (Pt) counter-electrode. Specific values of the experimental parameters were chosen, and the influence of the anodization time (from 0.2 s to 7 s) was investigated. The material properties of the liquid Ga nanoparticles were analyzed based on XRD measurements, and the morphology of the nanoparticles was characterized by SEM observation. The optical properties of the nanoparticles were characterized via the measurement of the diffuse reflectance spectra of the samples. It was found that the anodization induces the formation of a dimple-textured oxide layer at the surface of the nanoparticles. The process results in a red-shift of the plasmonic resonance.

The manuscript is well written, and the results are discussed in a clear and consistent way. I think that this manuscript can be published in Materials after minor revision. Here are some comments that will hopefully help the authors improving their manuscript before its publication:

  • Page 2, line 63: The authors should add some reference to support their statement about the Volmer-Weber growth mode. In the manuscript, the words “growth mode” are missing after “Volmer-Weber”.
  • Page 2, line 65: H3PO4 should be written with subscript numbers.
  • Page 3, line 96: The precision of the statement “peaks centered at 30º-50º” could be improved. For instance, the authors could add Gaussian fits to the graph shown in Fig. 1c to evidence the position of both peaks.
  • Page 3, line 114: Ga(OH)3 should be written with a subscript number.
  • Page 5, lines 193-194: The “red-shift in the resonance wavelength” should be quantified. The authors should at least provide some estimation. In Fig. 7b, the peak wavelengths of both resonances could be indicated for improving the clarity of the discussion.
  • Page 7, line 214: I think that “This electrochemically driven NP deformation” would be more correct.
  • I would like to know if the authors have considered the plasmonic interactions between the deposited nanoparticles. It seems that some quite narrow gaps can be observed on the SEM images shown in Fig 2. Does the anodization process modify the average width of the nanogaps between the nanoparticles? May it contribute to the red-shift of the plasmonic resonance? Would it be possible to estimate the enhancement of the localized electric field in such nanogaps based on numerical simulation results?
  • I would be grateful if the authors could add the following reference regarding the potential use of localized surface plasmon resonances in diverse fields:
    Fujiwara, H., Suzuki, T., Pin, C., & Sasaki, K. (2019). Localized ZnO growth on a gold nanoantenna by plasmon-assisted hydrothermal synthesis. Nano letters, 20(1), 389-394.

Reviewer 3 Report

Authors introduced a new method of plasmonic tuning of gallium nanoparticles by surface anodization in this paper. The phenomena were interesting, however, several issues should be resolved to be accepted.

  1. Even though the dimpled surface structure could induce significant red shift, there is no clear picture to be used for a future application, for example 'as sensor or some commensurate field'. It can be suggested in conclusion.
  2. Authors should provide information what happens with longer anodization time than 7 or 9 sec. What happens when the Ga NPs becomes full oxidation?
  3. Reviewer wondered if the Ga NPs on silicon wafer could be detached from the substrate.
  4. For "diffuse reflectance spectra" measurement, it should provide more detailed layout or methodology to measure LSPR. Even the UV-Vis-NIR was not described with its vendor or manufacturer.
  5. In Figure 1(c), even though the peaks of XRD were broad and shouldered, still the peaks can be assigned with respect to orderings. Was it complete amorphous? 
  6. Why did authors call or define Ga as "liquid" in this paper?
  7. In Figure 5, based on the reflectance profile, reviewer can imagine that different colored surfaces can be obtained under white light.
  8. Authors need to provide if there is further oxidation without anodization, under a very long period of exposure to air or oxygen. 
  9. In line 37~38, an abbreviation of SERS can be inserted. In Figure 4, there is a typo with "LSRP". It should be "LSPR'.

Reviewer 4 Report

Dear Editor,

Thank you for inviting me to review this manuscript titled “Plasmon tuning of liquid gallium nanoparticles through surface anodization”. In this paper, the authors developed a simple approach for tuning LSPR response of liquid Ga nanoparticles through surface anodization. The shape deformation of dimple textures was achieved on the Ga nanoparticles, which resulted in a red-shifting of the resonance wavelength. This original work provided preliminary study results of plasmon tuning based on deformation of liquid nanoparticles, yet the in-depth analysis and were not comprehensive enough. Therefore, I recommend a revision of this paper before publication in Materials.

Detailed comments listed below:

  1. Please provide the corresponding E-field distribution around the Ga nanoparticles in the FDTD simulation.
  2. Figure 4 showed the anodized oxide layer thickness of different samples, but there were no error bars on the measured LSPR shifts.
  3. Different morphologies (dimple depths) were recommended in the FDTD simulations of anodized Ga NPs.
  4. What is the theoretical basis for the red-shift of LSPR wavelength? The author described that the change in the particle shape is the dominant factor that affects the LSPR shift. Please provide enough evidence or references to support this claim.

Reviewer 5 Report

Report on the paper by Chen et al.

It is an interesting paper on the first report describing surface anodization of liquid Ga NPs. Thus I would like to recommend the publication of the paper after minor changes.

  1. An outline of the FDTD calculations for Figs. 6 and 7 should be described in the manuscript as well as the related references.

Round 2

Reviewer 1 Report

I thank the authors for having considered my suggestions and replied to each of my comments. I believe the current version of the manuscript has been substantially improved with respect to the previous.

Numerical and methodological details are now provided, the roles of dimpled surface versus the oxide layer thickness have been rephrased in a more clear manner, and the overall discussion of the results has been refined thanks to the significant effort of the authors.

I still have some minor comments which I believe, when addressed, will make the manuscript suitable for publication on Materials.

In particular:

  • Supporting Information: there are some misprints in the document, e.g. in the x-axis label of Fig. S2, caption of Fig. S3, y-label of Fig. S5 (as I doubt those are % data for reflection)
  • Figure 7a: the new labels “Depth: XX nm” in red are hard to see. I would suggest authors to consider to change colour. Trying with the same colour coding of Fig. 7b could be an idea, to make the figure even more readable.
  • Figure 7c,7d: are authors plotting the electric field enhancement, i.e. the solution of the electromagnetic simulation, normalised to the incident field? If so, this should be clearly stated in the figure caption, and a label should be also inserted next to the scale bar of the two panels.
  • Page 2 line 83: I would suggest authors to specify that simulation considers an ordered infinite array of NPs. Also, it may be useful to clarify that “nanogaps” discussed in Fig. S5 and relative comment in the main text amount for the in-plane periodicity of such a simulated array.
  • Page 2 line 86: are the authors actually monitoring transmission, or rather reflectance (the latter being the quantity shown in all the manuscript figures)?
  • Page 6 line 216: it remains a bit obscure to me the meaning of the sentence “their volume were evaluated by calculating volumes of intersections of two sphere of three spheres”. It is not clear why using two/three spheres, and computing volumes as intersections instead of integrals over the geometry, or as a difference, given the starting Ga NP volume and the removed material.
  • Page 8 line 259: in light of the new sentence suggesting potential applications of hot spots and LSPR from nanoporous surface to optical sensing, I point out the following references, that authors may want to consider to include in their conclusion paragraph: [ACS Appl. Mat. Inter. 11 (4), 3753-3762 (2019)], [J. Phys. Chem. C 123 (33), 20287-20296 (2919)].

Reviewer 3 Report

'Cause most of issues raised from reviewer were resolved. 

Author Response

Thank you.